# Pregnant Women’s Intentions to Deliver at a Health Facility in the Pastoralist Communities of Afar, Ethiopia: An Application of the Health Belief Model

**DOI:** 10.3390/ijerph16050888

**Published:** 2019-03-11

**Authors:** Znabu Hadush Kahsay, Molla Kahssay Hiluf, Reda Shamie, Yordanos Tadesse, Alessandra N. Bazzano

**Affiliations:** 1Health Education and Behavioral Science Unit, School of Public Health, Mekelle University, Mekelle 1871, Ethiopia; redashamie@yahoo.com (R.S.); Yodatad@gmail.com (Y.T.); 2Department of Public Health, College of Health Science, Samara University, Samara, Ethiopia; mollaka2005@gmail.com; 3Tulane University School of Public Health and Tropical Medicine, 1440 Canal Street, New Orleans, LA 70112, USA; abazzano@tulane.edu

**Keywords:** pregnant women, intention, health behavior, health belief model, Afar Region

## Abstract

Despite the significant benefits of giving birth at a health facility to improve maternal and child health, the practice remains lower than expected in pastoralist communities of Ethiopia. Understanding the intentions of pregnant women to use health facilities for delivery predicts the adoption of the behavior, yet documented evidence of intention in the context of pastoralist populations remains scarce. The current study aimed to assess pregnant women’s intentions to use a health facility for delivery in the Afar region of Ethiopia using the framework of the health belief model (HBM). A community-based, cross sectional survey was conducted from April 1 to April 30 2016 among 357 randomly sampled pregnant women using an interviewer-administered, semi-structured questionnaire. Data were entered into EpiData and exported to SPSS version 20.0 for analysis. Principal component factor analysis was done to extract relevant constructs of the model, and the reliability of items in each construct was assessed for acceptability. Multivariate logistic regressions were applied to identify predictors of pregnant women’s intentions to give birth at a health facility. The odds ratio was reported, and statistical significance was declared at 95% CI and 0.05 *p* value. Three hundred fifty seven pregnant women participated in the study (104.6% response rate indicating above the minimum sample size required). Among the respondents, only 108 (30.3%) participants intended to use a health facility for the delivery for their current pregnancy. Higher household average monthly income [AOR = 1.23, 95% CI = (1.10 − 2.90), antenatal clinic (ANC) attendance for their current pregnancy [AOR = 1.41, 95% CI = (1.31 − 2.10), perceived susceptibility to delivery-related complications [AOR = 1.52, 95% CI = (1.30 − 2.70), and perceived severity of the delivery complications [AOR = 1.66, 95% CI = (1.12 − 2.31) were positively associated with pregnant women’s intentions to deliver at a health facility. Intention was negatively associated with participants’ perceived barriers to accessing a health facility [AOR = 0.62, 95% CI = (0.36 − 0.85). **Conclusions**: A low proportion of pregnant women in the sampled community intended to deliver at a health facility. Pastoralist communities may have special needs in this regard, with household income, antenatal care attendance, perceived risk of complications, and perceived barriers to accessing a health facility largely explaining the variance in intention. Community-based interventions providing counseling and messaging on danger signs in the perinatal period and emphasizing benefits of delivering at a facility are recommended, alongside improving access.

## 1. Introduction

While maternal mortality has declined globally over recent decades, women continue to die of complications during pregnancy and childbirth. In 2015, 303,000 mothers died from such complications, with 99% of them from lower income countries (1). Furthermore, countries in Sub-Saharan Africa, including Ethiopia, account for a significant share of the global burden of maternal mortality ratio. The most recent maternal mortality ratio for Ethiopia was 353 deaths per 100,000 live births in 2015 [1,2].

However, evidence exists for giving birth at a health facility as a prominent strategy to reduce maternal and child mortality if quality emergency and obstetric care is in place by skilled attendants [3,4]. It could prevent 13% to 33% of maternal deaths, which may rise to 75% or more if care starts during labor and continues to the early postpartum period [4,5,6]. Recognizing the importance of encouraging facility-based birth, the Ethiopian government identified maternal and child survival as high priorities needing greater attention. The need to promote giving birth at health facility is reflected in many national health programs, including the most recent Health Sector Development Plans [7] and health sector transformation plan [8]. Despite strong ongoing efforts in the country, the national proportion of births assisted by skilled birth attendants is only 26%, with an even lower rate (16%) in the Afar pastoralist communities of Ethiopia [9].

Previous studies identified that health facility delivery is affected by individual level factors such as women’s knowledge, perception, attitudes, and individual beliefs regarding giving birth at a health facility [10,11,12,13,14,15]. Previous studies also highlighted that the likelihood of using health facilities for delivery is lower among women who have attained less education and who reside in rural areas [9,13,16,17,18]. 

Behavioral theories and models, such as the health belief model (HBM), facilitate an understanding of determinants of individual health-related behaviors and ways to stimulate positive changes. The health belief model (HBM) was developed by Becker in 1974 [19] and modified by Rostenstock in 1990 [20] as one of the value expectancy theories. The model postulates that the likelihood of behavior (e.g., delivery at health facility) is predicted by (1) the individual’s perceived threat towards the problem (severity of and susceptibility to the problem), (2) the perceived net benefit of adopting the behavior (if the perceived benefit outweighs the perceived barrier), (3) the individual’s perceived self-efficacy to perform the behavior, and (4) exposure towards cues-to-action (information that motivates adoption of the behavior) [21] 

Previous studies in pastoralist populations of Ethiopia have focused on the environmental and sociocultural factors of women, but have not done so utilizing the HBM as a framework to understand intention to deliver at a health facility. The current study aimed to determine pregnant women’s intention to give birth at a health facility and its associated factors in Afar pastoralist communities using the perspective of the health belief model (HBM). 

## 2. Methods and Materials 

### 2.1. Study Setting and Time

A community-based cross sectional study was conducted in Zone 3 (Gabi Rasu zone) of Afar region from April 1 to April 30, 2016. Zone 3 of Afar region is located 365 km from the south of Samara, the administrative city of Afar national regional state. The zone is bordered in the south by Oromia region, in the southwest by Amhara region, and in the east by Somali region. The total population of the zone was estimated to be 257,068 in 2016, 123,393 of whom were women residing in seven districts [22]. In terms of access to health facilities, the zone has only 1 primary hospital, 13 health centers, and 74 health posts (all of which are potential sites for health facility delivery). 

### 2.2. Sample Size and Sampling Procedure

Sample size was determined using single population proportion statistical formula with the following parameters: p = proportion of women who give birth at health facility in Afar region = 16% [9]; d = margin of error = 5%; confidence interval = 95%, and design effect = 1.5.
n = (Zα/2)^2^ ∗ P (1 − P)/d2 = 1.96^2^ ∗ 0.16(0.84)/(0.05)^2^ = 207

Then, considering a design effect of 1.5 i.e., 1.5 ∗ 207 = 310 and 10% contingency, i.e., 310 + (10% ∗ 310), the minimum sample size required for the study was 341. 

Twelve kebeles (the smallest administrative unit) from three randomly selected districts (Amibara, Gewane, and Argoba) were selected by lottery method. The sample size was proportionally allocated to each kebele based on the projected number of pregnant women. In each selected kebele, a list of pregnant women was developed to construct the sampling frame. Women who were at least 3 months gestational age (ascertained by self-report of last menstrual period) and who had lived in the study area at least for six months were included in the study. Finally, systematic random sampling was employed to select pregnant women from the list. Data collectors approached each selected pregnant woman at their home for an interview.

### 2.3. Measurement

A semi-structured questionnaire was adapted from validated examples in the literature, translated to the local language, and pretested on 10% of the sample size in a similar setting. The first section of the questionnaire contained sociodemographic and previous history of birth. The second consisted of items designed to assess respondents’ response to the constructs of the health belief model, namely (1) perceived susceptibility to birth-related complications, (2) perceived severity of the complications, (3) perceived barriers to delivery at a health facility, (4) perceived benefits of delivery at health facility, and (5) self-efficacy to use a health facility for delivery. For each item, respondents were asked their level of agreement or disagreement to items using a five-point Likert scale ranging from strongly agree (5) to strongly disagree (1). Consequently, 26 items were used to measure the constructs of the health belief model. 

The items were subjected to exploratory factor analysis with principal component analysis method, with a fixed number of constructs (i.e., five factors). Accordingly, the analysis identified perceived susceptibility (5 items), perceived severity (4 items), perceived benefits (5 items), perceived barriers (5 items), and perceived self-efficacy (2 items) as those to be extracted as constructs. All the extracted constructs explained jointly 52.1% of the variance of intention while perceived susceptibility alone explained 19.3% of the variation in the intention to use health facility for delivery.

Then, reliability testing of items in each construct was assessed before using the constructs for further analysis. The result of the test showed that Chronbanch’s α was above 0.70% for all constructs. For each construct, the items were summed up to produce a composite score, and the mean score was used for further analysis. 

**Outcome variable**: The outcome variable was intention to use a health facility for birth. It was measured by asking pregnant women about their plan for where they would deliver their baby for their current pregnancy. The women were asked to choose either home or a health facility. 

**Data Collection**: Trained diploma holder nurses who were fluent in local languages collected the data. The principal investigators trained data collectors and closely supervised the data collection process.

**Data management and analysis**: The data were entered into EpiData version 3.1 and then exported to SPSS version 20.0 for analysis. Descriptive statistics were used to summarize the results. The association between each independent variable and outcome variable was first assessed using binary logistic regression analysis. Variables with a *p* value of less than 0.05 were entered into multivariate logistic regression models. Adjusted odds ratios were reported at 95% confidence interval and a level of significance less than 0.05 was used to declare an association.

**Ethical considerations**: Ethical clearance was obtained from the ethical review committee of Samara University, Ethiopia. The purpose of the study was explained to all respondents, and written informed consent was obtained from each respondent after they were assured of its confidentiality. 

## 3. Results

### 3.1. Sociodemographic Characteristics of the Respondents

Three-hundred and fifty-seven pregnant women participated in the study, which was slightly more than the minimum sample size determined. The mean age of the respondents was 22 years old SD = ±4.7). In the study, 87.7% of the respondents were rural dwellers and 91.3% were Muslim in religion. Regarding ethnicity, the majority (80.4%) belonged to the Afar ethnic group. Meanwhile 74% of the respondents could read and write (see Table 1). 

### 3.2. Obstetric History of Respondents

With respect to obstetric history, 96% of the respondents were less than 19 years at first marriage. Although 80% of participants had ever used antenatal care services in their current pregnancy, only 34% obtained information about potential complications during birth while at an antenatal clinic (ANC), and 43% of them reported that they were not counseled about choosing a place of delivery. Nearly one in four reported an experience of abortion (of any form), while 35% reported a history of birth complications. 

Of those respondents who had ever given birth (*n* = 302), only 58 (19.2%) delivered their index child in a health facility (Table 2). 

### 3.3. Intention of Pregnant Women to Use Health Facility for Birth 

Of the participants, only 30.3% of the participants intended/planned to use a health facility for delivery for their current pregnancy (Table 2).

The mean score of respondents was computed for each construct of the health belief model, which was framed around health facility use for delivery. Consequently, the mean scores were 13.6 ± 2.6 (range of possible values: 5–25) for perceived susceptibility, 17.0 ± 3.7 (range of possible values: 5–25) for perceived severity, 15.5 ± 3.3 (range of possible values: 4–20) for perceived barriers, 11.5 ± 3.6 (range of possible values: 5–25) for perceived benefit, and 6.6 ± 2.3 (range of possible values: 2–10) for self-efficacy.

The relationship between intention to give birth at a health facility and predictor variables was first assessed through bivariate analysis. In this analysis, household income, age at first marriage, experience of antenatal care visit, being informed about delivery complications, experience of delivery complications, perceived susceptibility, perceived severity, and perceived benefit were statistically significantly associated with intention to use a health facility for delivery (*p* < 0.05). 

To examine the independent effect of each predictor variable, multivariate logistic regression was performed, and the result is displayed in Table 3. Consequently, women with a household monthly income of ≥500 ETB were 1.23 times more likely to intend to use a health facility for birth compared to women with <500 ETB. Similarly, women who attended an ANC visit for their current pregnancy were 1.41 times more likely to intend to use a health facility for delivery than their counterparts. Of the health belief model constructs, a unit increases in score of perceived susceptibility and perceived severity was found to increase the probability of intending to use health facility for delivery (significantly) by 1.52 times and 1.66 times, respectively. Additionally, perceived barriers were negatively associated with intention to use health facility for delivery. A unit increase in score of perceived barriers was found to reduce the probability of intending to use a health facility for delivery by 38% (Table 3). 

## 4. Discussion

This study assessed pregnant women’s intention to use a health facility for birth in pastoralist communities of Afar region in Ethiopia using the perspective of the health belief model. Among 302 women who have ever given birth, 58 (19.2%) gave their last baby in a health facility. This finding was similar to the 16% proportion that the Ethiopian Demographic and Health Survey found for Afar region in 2016 [9]. 

According to some behavioral change theories, individual behavior is driven by behavioral intention and the degree to which people are expected to carry out their intention when the opportunity arises. Intention is thus assumed to be the immediate antecedent of behavior [23]. In this study, about 30.3% of the respondents intended to give birth in health facility for their current pregnancy. This result is higher compared with previous studies from this region, in which 18.9% of expecting women intended to use a health facility to deliver their babies [21]. This might be due to temporality differences and increased interventions to promote health facility use for delivery in recent times in Ethiopia, or to improvements in access skilled birth attendants. However, another study from the region also found a lower level of intention to use a health facility [23].

Pregnant women’s perceived susceptibility and increased perceived severity towards birth-related complications were significantly associated with their intention to use a health facility for birth. Previous studies also show low awareness regarding the possible birth complications impacting the decision to use a health facility for delivery [2,10,13]. Pregnant women’s level of perception regarding birth complications could be determined based on their previous experience with themselves or with others they have seen [24]. In addition, the effect of previous birth experiences is recognized as a determinant for subsequent delivery locations [24]. Therefore, mothers may prefer giving birth at home based on past successful experiences with homebirths and the possible positive attitude towards traditional birth attendants regarded as respectful and friendly during assistance [25]. In this regard, nearly one in four of the women participated in the current study reported the experience of abortion (of any form), while 35% of them reported history of birth complications, which is in tandem with a previous study [14]. This may imply heightening women’s perception regarding the susceptibility and severity of pregnancy and birth-related complications could encourage health facility use for birth. 

Perceived barriers to accessing a health facility were negatively associated with intending to use a health facility for birth. Perceived cost, distance, and inconvenience were among the items included in the perceived barrier construct, indicating the items explain substantial variation in women’s decisions on the place to give birth. Previous studies also illustrated that women’s perception regarding the barriers to accessing facilities for delivery, such as inability to pay for the costs related to the services, distance, and transportation [10,24,25,26,27,28], are important. Likewise, a previous study also showed that perceived barriers were strong determinants of intention to give birth in health facility [23,29]. Long distance, costs related to travelling to a health facility, and the feeling of being ashamed to be naked for services influence pregnant women’s decision on birth place [24,25,30]. Provision of access to transportation and strengthening counseling for birth preparedness may be promising ways to reduce the barriers women perceive. 

Regarding ANC service use, 283 (79.3%) had ever used antenatal care services during their current pregnancy, which is a higher figure compared to previous estimates from Afar region [9] and elsewhere [21,31]. Nevertheless, there were also earlier studies indicating higher rates of ANC utilization [32,33]. However, among women who attended an ANC program, only 43% were provided information about possible complications during delivery and childbirth. Lack of information about where to deliver suggests the existence of a significant missed opportunity for education and counseling services. A previous study noted that a large proportion of women who attended an ANC program obtained better information on possible complications (72.8%) and were also counseled on the place of delivery (77%) [32]. The finding identifies the need to provide focused and individualized education and counseling to each pregnant woman who come in contact with a health facility about the benefits of delivering in a health facility, the potential risks and complications associated with pregnancy in general, and home delivery in particular. 

## 5. Conclusions and Recommendations

The majority of pregnant women in the current study in the pastoralist community of Afar were not intending to deliver at a health facility for their current pregnancy. Perceived susceptibility to and severity of delivery-related complications were associated with an intention to use a health facility for delivery, while perceived barriers were negatively correlated. Strengthening health education with counseling on the danger signs and delivery-related complications during antenatal care counseling and awareness creation activities are recommended. In addition, tailored health interventions should be targeted to reduce barriers to use of health facilities. 

## 6. Limitation of the Study

This study applied the health belief model, which emphasizes individual level factors. Thus, other external determinants such as social and structural factors may not have been included in the findings. 

## Figures and Tables

**Table 1 ijerph-16-00888-t001:** Pregnant women’s socio-demographic characteristics in Gabi-Rasu zone of Afar region, Ethiopia, 2016 (*n* = 357).

Variables	Categories	Frequency	Percent
Residence	Rural	313	87.7
	Urban	44	12.3
Age in years	≤24	117	32.7
	25–29	127	35.6
	≥30	113	31.7
Religion	Muslim	326	91.3
	Orthodox	25	7.0
	Other *	6	1.7
Ethnicity	Afar	287	80.4
	Argoba	38	10.6
	Amhara	25	7.0
	Other **	7	2.0
Educational Status	Illiterate	263	73.7
	1–4	91	25.5
	≥5	3	0.6
Occupational Status	House Wife	232	64.4
	Merchant	55	15.4
	Employed	36	10.1
	Farmer	34	9.5
Current Marital status	Married	333	93.3
	Unmarried	12	3.4
	Other ***	12	3.3
Average household monthly Income [Eth Birr]	<500	178	49.9
	≥500	179	50.1

* = Protestant and Catholic ** = Oromo, Wolayta and Hadiya *** = Widowed and Divorced.

**Table 2 ijerph-16-00888-t002:** Pregnant women’s obstetric characteristics in Gabi-Rasu zone of Afar region, Ethiopia, 2016 (*n* = 357).

Characteristics	Category	Frequency	Percentage (%)
Age at first marriage	≤19	341	95.5
	>19	16	4.5
Number of pregnancies including the current	<5	280	78.4
	≥5	75	21.9
ANC attendance during current pregnancy	Yes	283	79.3
	No	74	20.7
Informed about delivery complications	Yes	122	34.1
	No	161	49.1
Informed about place of delivery	Yes	138	38.7
	No	145	43.2
History of abortion (of any type)	Yes	81	22.7
	No	276	77.3
Ever had birth related complication	Yes	125	35.0
	No	217	64.0
Place of delivery of for index child	Home	244	76.8
	Health facility	58	19.2
Delivery attendants for index child	TBAs	136	45.0
	Relatives/friends/neighbors	61	20.2
	Health professionals	58	19.2
	HEWs	47	15.5
Plan for place of delivery for current pregnancy	Home	249	69.7
	Health facility	108	30.3

ANC—Antenatal care, TBA = traditional birth attendant, HEW = health extension worker.

**Table 3 ijerph-16-00888-t003:** Factors associated with pregnant women’s intention to use a health facility for delivery, Gabi-Rasu zone, Afar region, Ethiopia, 2016.

Factors		Crude Odds Ratio (COR) (95% CI)	B	Adjusted Odds Ratio (AOR) (95% CI)
Average monthly income [Eth Birr]	≥500	1.92 (1.21–3.05)	0.25	1.23 (1.10–2.90)
	<500 (ref) *	1		1
Age at first marriage [in year]	≤19	1.39 (1.21–1.57)	0.22	1.87 (1.02–2.38)
	>19 (ref)	1		1
Ever used antenatal care for the current pregnancy	Yes	2.56 (1.98–4.68)	0.77	1.41 (1.31–2.10)
	No (ref)	1		1
Perceived Susceptibility	Mean = 13.6±2.6	1.07 (0.99–1.15)	0.03	1.52 (1.30–2.70)
Perceived Severity	Mean = 17.0 ± 3.7	1.16 (1.07–1.25)	0.14	1.66 (1.12–2.31)
Perceived Barrier	Mean = 15.5 ± 3.3	0.92 (0.86–0.98)	−0.07	0.62 (0.36–0.85)

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
