# Peer review of "Pregnant Women’s Intentions to Deliver at a Health Facility in the Pastoralist Communities of Afar, Ethiopia: An Application of the Health Belief Model"

_ijerph, 2019, doi:10.3390/ijerph16050888_

Round 1

Reviewer 1 Report

This work is well written and deals with a topic of profound interest in the health of women in a developing Country, where many factors can contribute to a high infant mortality rate. 

The work is dealt with in its different sections in a scientific and up-to-date manner and the epidemiological statistical aspect is well reported.

I think it is useful that this work can also be published to represent indicators to be pursued to improve the health of women in such an important moment as pregnancy and the choice to give birth  in a health facility with safety standards and requirements for the unborn child.

Author Response

Our great appreciation to the reviewers for spending time to oversee the relevance of our contribution to the readers of IJERPH. 

Reviewer 2 Report

Abstract:

In the methods section sample size 341 and results section sample size 357. Please clear it. 

In the introduction, sample size ,207,310, 341. Please clear it. 

Result section: Define AOR, ANC

Introduction:

Text in Line 57: 59 is not clear

 Line: 8-88: Sample size section with  design effect and 10% contingency is not clear

Line 109: you mentioned principal component analysis. You need to show the result in table format. 

Line: 139: Did you accidentally delete Majoriy? 

Line 192: Is it multiple regression or multivariate regression and same for line 21 also

How much variability does explain by HBM? You can show number as you did regression analysis.

In table 2: something wrong with column 4

In table 3; Add SD on perceived susceptibility, severity and barrier

Also correct decimal up 2 or 3, example 0.253, somewhere 0.22

Author Response

We would like to appreciate the the reviewer for the careful assessment of our paper and the provided comments here. The comments and suggestions really helped us to improve the manuscript further. We hoped the point by point responses provided here within would be upon the satisfaction of the reviewers.

Best regards! 

Abstract:

Point 1: In the methods section sample size 341 and results section sample size 357. Please clear it. 

Response 1: Thank you for point this type error: We determined the minimum sample size required (341) using a single population formula (207). Then we multiplied it with a design effect of 1.5 to minimize sampling error (310). Finally, we added a 10% contingency for possible missing values. But, fortunately, we approached 357 pregnant women, which is above the minimum sample size determined. We didn’t preferred to drop the 17 participants because the data was collected, yielding the final number of participants 157. To make this process clear for readers, we paraphrased the sentence in the methods section as

“Sample size was determined using single population proportion statistical formula with the following parameters; p=Proportion of women who give birth at health facility in Afar region=16% [9]; d=Margin of error=5%) Confidence interval=95% and design effect = 1.5.

n= (Zα/2)2 *P (1-P) /d2 = 1.962 * 0.16(0.84)/ (0.05)2 = 207

Then considering a design effect of 1.5 i.e. 1.5%*207=310 and 10% contingency, i.e. 310+(10%*310) the minimum sample size required for the study was 341.”

Point 2: In the introduction, sample size ,207,310, 341. Please clear it. 

Response 2: Already addressed above

Point 3: Result section: Define AOR, ANC:

Response 3: Thank you for raising an important issue: now we provided the full name as Adjusted odds ratio (AOR) and antenatal care (ANC).

Point 4: Introduction: Text in Line 57: 59 is not clear:

Response 4: The paragraph is rewritten as: Previous studies identified that health facility delivery is affected by individual level factors such as women’s knowledge, perception, attitudes, and individual beliefs regarding giving birth at health facility [10-15]. Previous studies also highlighted that likelihood of using health facilities for delivery is lower among women who have attained less education, and who reside in rural areas [9, 13, 16-18].

 Point 5: Line: 8-88: Sample size section with design effect and 10% contingency is not clear

Response 5: We paraphrased it as, “Then considering a design effect of 1.5 i.e. 1.5%*207=310 and 10% contingency, i.e. 310+ (10%*310) the minimum sample size required for the study was 341”.  Then, if the modification is not adequate, we will look to hear from you.   

Point 6: Line 109: you mentioned principal component analysis. You need to show the result in table format. 

Point 7: Line: 139: Did you accidentally delete Majoriy? 

Response 7: Of course yes. Now it appears at it was done.

Point 8: Line 192: Is it multiple regression or multivariate regression and same for line 21 also

Response 8: In all sections, we have changed multiple regression to multivariate regression, which seems appropriate.

Point 9: How much variability does explain by HBM? You can show number as you did regression analysis.

Response 8: We have provided the total variation of intention to use HF for delivery in the methods section, which 52%. If it needs to be in the result section, we wanted to hear from you. Unless, if could be comfortable for readers it is provided in the measurement section of the method along with other detailed description of the components of the HBM.

Point 9: In table 2: something wrong with column 4

Response 8: Thank you the labeling of the column was missed. We provided the labeling now.

Point10: In table 3; Add SD on perceived susceptibility, severity and barrier

Response 8: Now we have provided the SD for the means scores in table 3

Point 11: Also correct decimal up 2 or 3, example 0.253, somewhere 0.22

Response 11: Thank you again. It is now done